# Predicting the Risk of Recurrent Venous Thromboembolism: Current Challenges and Future Opportunities

**DOI:** 10.3390/jcm9051582

**Published:** 2020-05-22

**Authors:** Hannah Stevens, Karlheinz Peter, Huyen Tran, James McFadyen

**Affiliations:** 1Clinical Haematology Department, Alfred Hospital, Melbourne, VIC 3004, Australia; huyen.tran@monash.edu (H.T.); james.mcfadyen@monash.edu (J.M.); 2Australian Centre for Blood Diseases, Monash University, Melbourne, VIC 3004, Australia; 3Atherothrombosis and Vascular Biology Program, Baker Heart and Diabetes Institute, Melbourne, VIC 3004, Australia; Karlheinz.Peter@baker.edu.au; 4Department of Cardiology, Alfred Hospital, Melbourne, VIC 3004, Australia

**Keywords:** venous thromboembolism, pulmonary embolism, deep vein thrombosis, biomarker, risk stratification

## Abstract

Acute venous thromboembolism (VTE) is a commonly diagnosed condition and requires treatment with anticoagulation to reduce the risk of embolisation as well as recurrent venous thrombotic events. In many cases, cessation of anticoagulation is associated with an unacceptably high risk of recurrent VTE, precipitating the use of indefinite anticoagulation. In contrast, however, continuing anticoagulation is associated with increased major bleeding events. As a consequence, it is essential to accurately predict the subgroup of patients who have the highest probability of experiencing recurrent VTE, so that treatment can be appropriately tailored to each individual. To this end, the development of clinical prediction models has aided in calculating the risk of recurrent thrombotic events; however, there are several limitations with regards to routine use for all patients with acute VTE. More recently, focus has shifted towards the utility of novel biomarkers in the understanding of disease pathogenesis as well as their application in predicting recurrent VTE. Below, we review the current strategies used to predict the development of recurrent VTE, with emphasis on the application of several promising novel biomarkers in this field.

## 1. Introduction

Venous thromboembolism (VTE) is a term that encompasses the diagnoses of both deep vein thrombosis (DVT) and pulmonary embolism (PE). VTE is associated with a significant global burden of disease with an estimated incidence of 0.5–2 per 1000 individuals in the general population and increases significantly to 2–7 per 1000 in those more than 70 years of age [1].

Acute VTE can be successfully prevented and treated with anticoagulant therapy; however, this comes with the inherent risk of bleeding, which in many cases may offset the clinical benefit of anticoagulation. With this in mind, new tools are required to accurately stratify patients who are at risk of recurrent VTE and who stand to benefit from ongoing anticoagulant therapy. As such, it is important to utilise our current understanding of the molecular underpinnings of venous thrombus formation and assess if there are novel tools, or biomarkers, which may aid in the prediction of recurrent VTE.

## 2. Treatment and Prediction of Recurrent Venous Thromboembolism

Treatment of VTE primarily involves therapeutic anticoagulation with a direct oral anticoagulant or vitamin K antagonist. In general, guidelines recommended at least three months of therapeutic anticoagulation for an acute VTE and extended duration anticoagulation largely depends on the provoking risk factor and the subsequent risk of recurrent VTE [2,3,4]. In this regard, it has been established that patients with VTE provoked by major surgery or trauma have a low risk of recurrence (approximately 3% at five years). In contrast, an unprovoked VTE is associated with a higher risk of recurrence (approximately 25–30% at five years), so that indefinite anticoagulation is often recommended by clinicians [5,6]. In some cases, the risk of recurrence is offset by the risk of bleeding, leading to clinical equipoise with regards to the most appropriate duration of treatment for many patients, particularly those with unprovoked VTE. As such, the use of clinical prediction tools or biomarkers is required to ensure that patients are appropriately risk-stratified and either continue anticoagulation due to high risk of recurrent VTE or have therapy appropriately ceased. This review will discuss the utility of some of the most pertinent prognostic aids and biomarkers used to assist in predicting risk of recurrent venous thrombosis, in addition to providing an overview of emerging technologies that may further enhance our ability to accurately risk-stratify patients.

## 3. Clinical Risk Prediction Models

The aim of utilising a clinical prediction model in VTE is to allow the risk stratification of patients with VTE treated with anticoagulation into those who have a low risk of recurrent VTE and thus can cease anticoagulation, and those patients who are at a high risk of recurrent VTE and should be recommended for continuing therapy. Several prediction models have been developed which aim to predict the likelihood of recurrent thrombotic events following an acute unprovoked VTE, and these include the HERDOO2 score [7,8], Vienna prediction model [9,10], and the DASH score [11,12]. A comparison of each of these models is shown in Table 1. Recently, an additional prediction model, the Leiden Thrombosis Recurrence Risk Prediction (L-TRRiP) model, has been evaluated and shows some promising data in this area but still requires external validation [13].

However, several limitations with the available clinical prediction models have been identified, including the definition of unprovoked VTE, with both the DASH and HERDOO2 studies incorporating hormone-associated VTE into the definition of an unprovoked index VTE. Additionally, there is a significant proportion of patients who are misclassified as having a low risk of recurrent VTE, and thus improved sensitivity across all models is required [14,15]. Furthermore, not all trials support the utility of the available prediction models to accurately risk-stratify patients at low and high risk of recurrent VTE, particularly if local laboratory measurement of D-dimer is included [16,17]. However, despite these limitations, the models discussed have identified factors that have consistently proven valuable in improving the prediction of recurrent VTE. Moving forward, it will be important to build on these foundations and consider the incorporation of biomarkers or radiological findings in combination with these traditional clinical characteristics to improve accuracy in predicting the risk of recurrent VTE.

## 4. The Utility of Imaging in Predicting Recurrent VTE

Following treatment for acute VTE, assessment of residual venous obstruction (RVO) post-DVT or residual pulmonary obstruction (RPO) may be obtained. RVO is primarily assessed by compression ultrasonography (CUS), whilst RPO can be evaluated using either lung scintigraphy (VQ scan) or computed tomography pulmonary angiography (CTPA).

### 4.1. Residual Venous Obstruction Following Deep Vein Thrombosis

The potential importance of RVO post-DVT has been studied for many years, with the concept of tailoring anticoagulation to RVO having been evaluated in several clinical trials [18,19]. To date, published systematic reviews and meta-analyses demonstrate a modest increase in the risk of recurrent VTE if RVO is present [20,21,22,23]. Importantly, this increased risk does not appear to translate to patients with unprovoked DVT [20,23]; the group of patients where uncertainty regarding the duration of anticoagulation exists and who would benefit most from a tool to predict recurrent thrombosis. Of note, the one study population where RVO appears to be associated with recurrent VTE is in patients with malignancy-associated VTE, and RVO assessment may be useful when considering optimal duration of anticoagulation in this patient cohort, but the overall number of patients evaluated has been small [21,23].

An emerging imaging modality for evaluating DVT is magnetic resonance direct thrombus imaging (MRDTI), which appears to be particularly useful at diagnosing or excluding acute, recurrent DVT in the setting of chronic DVT. MRDTI requires no intravenous contrast and the image is based on the formation of methemoglobin in an acute thrombus, which is visible as a high signal on T1-weighted sequence on magnetic resonance imaging (MRI) [24,25]. Evaluation of recurrent DVT in the setting of a chronic thrombus may lead to an inconclusive result when imaged with CUS, and thus MRDTI is an attractive option for evaluating suspected recurrent ipsilateral DVT, as this modality can improve accuracy in delineating between acute and chronic thrombus [24]. These promising results lead to speculation that MRDTI may also be beneficial in risk prediction for recurrent VTE, but MRDTI is yet to be evaluated for this purpose.

### 4.2. Residual Pulmonary Obstruction

The rate of RPO following an acute PE appears to depend on the size of the initial clot burden and is reported to be as high as 60% [26,27]. Many studies have now evaluated the utility of RPO in predicting recurrent PE in patients with an acute PE treated with at least three months of oral anticoagulation. The findings of these studies are mixed, with several analyses concluding that there is a strong link between RPO and recurrent PE [28,29,30,31] and many also inferring that there are no associations [32,33,34,35]. When comparing these studies, it must be noted that imaging techniques are not standardised and include both VQ scan and CTPA as well as a variety of time points when repeat imaging was performed. Importantly, a significant proportion of patients in all studies continued on long-term anticoagulation, which impacts the natural history of the disease and alters the primary outcome of recurrent VTE.

The largest multicentre prospective study in this area, which reported on the outcomes of 647 patients with first episode of acute symptomatic PE, demonstrated an association between RPO, evaluated with VQ scan, and recurrent VTE as well as an increased risk of chronic thromboembolic pulmonary hypertension (CTEPH). Pesavento et al. showed that, in patients with RPO, 25/324 (7.7%) developed recurrent VTE versus 15/323 (4.6%) in patients without RPO, and RPO was found to be an independent predictor of both VTE recurrence and CTEPH (hazard ratio (HR) 2.26, 95% confidence interval (CI) 1.23–4.16, *p* = 0.009) [29]. Furthermore, a recent meta-analysis evaluating RPO and recurrent VTE determined that RPO was associated with an increased risk of recurrent VTE detected by VQ scan (odds ratio (OR) 2.22, 95% CI 1.61–3.05) [26]. These findings demonstrate that RPO detection following acute PE may increase the risk of recurrent VTE, but as the overall rate of recurrence in this cohort remains relatively low, its utility in predicting the likelihood of recurrent events is minimal, and it should not be used primarily for this purpose.

In summary, assessment of RVO post-DVT and RPO may confer a slight increase in recurrent VTE, but their utility in predicting recurrent VTE remains low, and thus their use is not recommended in routine clinical care.

## 5. Biomarkers in Venous Thromboembolism

Currently, there remains significant scope to broaden the standard of care that is used to predict recurrent VTE. In theory, novel biomarkers offer an alternative solution, by leveraging our current knowledge of the pathogenesis of VTE to predict the likelihood of recurrence. In an era that is moving toward personalised medicine, using biomarkers that consider the individual biological response to a disease process appears essential for improving patient outcomes. Below, we discuss some of the most widely studied biomarkers and their value in predicting recurrent venous thrombosis.

### 5.1. D-Dimer

D-dimer is a fibrin degradation product that is released when cross-linked fibrin is cleaved by plasmin. It has a high negative predictive value in the setting of acute VTE but can also be elevated in many other conditions, including cardiovascular disease, malignancy, infection, and pregnancy [36,37]. D-dimer is most commonly used in conjunction with a clinical prediction tool such as the Wells score or recently described YEARS score to help exclude VTE without the requirement for radiological imaging [38,39,40,41]. Due to its utility in acute VTE, D-dimer assessment has also been evaluated in predicting recurrent VTE. The PROLONG Study was the largest multicentre trial evaluating the use of D-dimer. This study assessed patients with a first episode of symptomatic, unprovoked VTE, who had completed three months of therapeutic anticoagulation with D-dimer testing performed 30 days after cessation of treatment. The final analysis demonstrated that an abnormal D-dimer was associated with a higher risk of recurrent VTE than in patients where the D-dimer was normal (15.0% versus 6.2%, respectively, *p* = 0.003) [42]. Similarly, the PROLONG II study prospectively evaluated the utility of multiple D-dimer measurements in patients with a first unprovoked VTE and assessed for recurrent VTE in this group. Again, the study demonstrated that the rate of recurrence is significantly higher in patients with an abnormal D-dimer when compared with patients with a normal D-dimer [43]. However, plasma D-dimer testing following an unprovoked VTE as the sole test for predicting recurrent VTE is not commonly performed as D-dimer alone cannot adequately distinguish which patient will develop a recurrence. Importantly, when using the data from the PROLONG study, even if the D-dimer is negative following treatment, one in 20 patients will still develop recurrent VTE. Accordingly, the most recent American College of Chest Physicians (CHEST) guidelines noted that a female with an unprovoked VTE and a negative posttreatment D-dimer might have a reduced risk of recurrence (similar to that of a nonsurgical risk factor), but a negative posttreatment D-dimer does not change risk profiles in males [3]. Despite these challenges, D-dimer is an easily accessible laboratory test and can result in a high negative predictive value in females for excluding VTE recurrence, particularly if incorporated in a clinical prediction rule.

### 5.2. Coagulation Factors

In the setting of VTE, the predictive value of coagulation factors has been of interest; however, only factor (F) VIII has been shown to have predictive value for recurrent VTE. Multiple studies have demonstrated that an elevated FVIII is associated with an increased risk of recurrent venous thrombosis, but, to date, not all results support the predictive role of this factor [44,45,46,47]. Interestingly, it has been established that, in the context of VTE, high levels of FVIII may persist over time and thus are not simply attributable to an acute phase reaction [48,49]. Recently, elevated FVIII levels were found to be associated with recurrent venous thrombosis following a first unprovoked VTE as well as following a first provoked event. Additionally, combining the measurement of FVIII to a clinical prediction tool, the DASH score, results in an improved predictive value of the score and supports the notion that the measurement of FVIII levels may be useful in aiding to predict recurrent VTE [47]. However, elevated FVIII alone is not a sensitive tool for predicting recurrent VTE but could be useful when combined with other clinical factors or novel biomarkers.

### 5.3. P-Selectin

P-selectin (CD62P, GMP-140) is a member of the selectin family of adhesion molecules and plays a central role in promoting leucocyte adhesion and recruitment at the site of vascular injury and during inflammation [50,51]. P-selectin is stored in the alpha granules of platelets and in Weibel-Palade bodies of endothelial cells, where upon activation, P-selectin is released such that it can bind with its cognate ligand, P-selectin glycoprotein ligand 1 (PSGL-1) found on leucocytes, including monocytes and neutrophils [50]. As P-selectin appears to be crucial for platelet-leucocyte and endothelial-leucocyte interactions, it has been evaluated for its role in the pathogenesis of VTE. Indeed, animal models have confirmed that P-selectin is required for leucocyte accumulation and fibrin deposition in VTE formation [52]. Accordingly, P-selectin-deficient mice are demonstrated to be protected from DVT development [53], confirming the importance of this molecule in the early stages of venous thrombosis formation.

In addition to the membrane form of P-selectin, a soluble form of the molecule has been detected in the plasma of mice and humans, and is aptly referred to as soluble P-selectin (sP-selectin). It is postulated that sP-selectin originates as either an alternatively spliced protein lacking a transmembrane domain, or cleavage of the membrane form of the molecule [54,55]. Soluble P-selectin can be readily detected in human plasma by way of enzyme-linked immunosorbent assay, which has made it a convenient surrogate marker of P-selectin expression, and its application has been evaluated in a range of disease states with a focus on arterial and venous thrombotic disorders [56,57,58].

The use of sP-selectin as a biomarker for VTE has been extensively evaluated at different time points during the disease state. There is general agreement that sP-selectin is increased in the setting of acute DVT or PE and it has been evaluated alone and in combination with other biomarkers to predict risk of acute VTE. Ramacciotti et al. demonstrated that sP-selectin in combination with a Wells score ≥ 2 was an effective tool for confirming DVT, but this approach has not been widely adopted and in the majority of patients would not negate the requirement for ultrasonography for diagnosis [59]. Additionally, sP-selectin has been shown to be associated with risk of recurrent VTE, with Kyrle et al. finding that the cumulative probability of VTE was significantly higher with P-selectin values above the 75th centile as compared with patients with lower levels (four-year cumulative recurrence 20.6% versus 10.8%, respectively). However, these studies demonstrate that a substantial proportion of patients develop recurrent VTE without a parallel rise in sP-selectin and, as such, it can be concluded that sP-selectin lacks sensitivity for recurrent VTE and thus cannot be used as a sole biomarker to predict recurrence.

### 5.4. Endothelial Progenitor Cells

Endothelial cell function is crucial to maintaining vascular integrity and homeostasis. More recently, the important role of the endothelium in inducing venous thrombosis formation is becoming increasingly recognised. In this regard, endothelial cells respond to injury or insult by downregulating anticoagulant proteins, increasing von Willebrand factor (vWF) expression, upregulating adhesive proteins such as P-selectin and E-selectin, and inducing inflammatory cytokine signaling; all of which aids in subsequent leucocyte and platelet recruitment [60]. Additionally, it is thought that signaling mechanisms in the setting of inflammation can mobilise endothelial progenitor cells (EPCs) to the site of vessel injury to aid in vascular regeneration [61]. Therefore, it has been postulated that the measurement of EPCs could be an effective marker of vascular regeneration after endothelial injury or damage, with several studies indicating that cardiovascular risk factors, such as diabetes, may adversely affect EPC number. Accordingly, a change in EPC number, or function, appears to be associated with poor cardiovascular outcomes [62,63]. Intriguingly, there is growing evidence from murine venous thrombosis models that EPCs may be important in thrombus resolution [64,65]. Recently, a post hoc analysis of the ExACT study, a multicentre randomised control trial comparing extended anticoagulation with discontinuation of anticoagulation following a first unprovoked VTE, demonstrated that patients with recurrent VTE had significantly lower levels of circulating EPCs [66]. Thus, further research of the role of EPCs in VTE pathogenesis in addition to larger datasets regarding their utility in predicting recurrent VTE will be welcomed.

### 5.5. Microvesicles

Microvesicles (MVs) are small membrane vesicles ranging between 100 and 1000 nm in diameter that are released from virtually all eukaryotic cells in the setting of cell activation or apoptosis [67,68]. Platelet-derived MVs are the predominant form of MVs found in human plasma and were first described in 1967 by Wolf as ‘platelet dust’ [69]. Since then, a key role for MVs in cell signalling and communication has been proposed, which is underscored by experimental evidence pointing to the role of MVs in the pathogenesis of many disease states, including thrombosis, malignancy, cardiovascular disease, and infection [70,71,72,73,74].

The pathogenic role of MVs in VTE has been suggested to occur via several mechanisms. The first is increased exposure of membrane phospholipids, such as the negatively charged phosphatidylserine (PS), which provides the requisite negatively charged surface for the assembly of the tenase and prothrombinase complexes required for efficient thrombin generation [67,71,75]. Secondly, MVs from some cell types express tissue factor (TF), which is a potent activator of coagulation [67]. These TF-positive MVs are thought to be predominantly derived from activated monocytes [73], with monocyte MVs believed to promote thrombin generation and trigger fibrin formation via the TF-dependent pathway [76]. Additionally, TF-positive MVs can also fuse with activated platelets and propagate the coagulation cascade [77].

Given that evidence points towards the importance of MVs in thrombus formation at a molecular level, their utility as a biomarker in VTE has been evaluated. Several studies have evaluated MVs in acute VTE and have found an association between elevated plasma MVs and acute VTE [78,79,80]; however, it is noteworthy that one study found that this association was no longer significant when adjusted for cardiovascular risk factors [78]. Interestingly, patients with inherited thrombophilia, such as antithrombin deficiency and Protein C deficiency, also exhibit elevated numbers of MVs [81]. Additionally, Ye et al. described an increase in TF-positive MVs in acute recurrent VTE but no association with a primary VTE [82]. In contrast, there was no association between MVs and VTE among patients with a prior history of VTE with MVs evaluated more than three months from the acute event [83]. However, it is important to recognise that these studies evaluate different types of MVs, including endothelial cell MVs, platelet MVs, and TF-positive MVs, and the isolation protocols as well as the markers used to identify MVs also varied between studies. Due to the lack of standardisation, any direct comparison between studies is challenging, and this should be considered when interpreting results.

Despite some promising data in this research area, there are several issues to be addressed within this field of research. Firstly, a lack of standardisation between laboratories when enumerating MV has been an ongoing concern. For example, the type of anticoagulant used for blood collection, isolation protocol, storage, and repertoire of antigens used to define the cellular origin are all variables which can lead to variation in the characterisation of MVs between laboratories [84]. In addition, the analysis of MVs continues to advance with the incorporation of flow cytometry, nanoparticle tracking analysis, and single-cell technology. It is also important to note that other extracellular vesicles, such as exosomes and apoptotic bodies, may overlap in size with MVs, something that has not previously been investigated in the field of VTE. In an attempt to try and harmonise methodology, nomenclature, and analysis of MVs, the International Society for Thrombosis and Haemostasis has published guidelines to enhance the reproducibility of MV research [85,86].

Therefore, to date, the role of MVs as a biomarker in VTE remains preliminary, and studies evaluating their utility in predicting recurrence VTE are lacking. However, with the standardisation and rapid technological advancements allowing characterisation of the biological and physical properties of MVs, this exciting field of research is hoped to yield insights in VTE pathogenesis, and in turn may define further novel biomarkers that can be used to help predict VTE risk.

### 5.6. C-Reactive Protein

The enhanced understanding of the molecular foundations of both arterial and venous thrombosis has highlighted that thrombus formation is an inflammatory condition. This has led to interest in the evaluation of laboratory markers of inflammation as a means to help predict VTE recurrence. One marker that has received significant interest is C-Reactive Protein (CRP), which is commonly used in clinical practice as a marker of inflammation. Interestingly, there is a growing body of evidence that, in addition to being a marker of inflammation, CRP plays an important role in mediating inflammatory and thrombotic reactions [87]. In this regard, it is now well established that CRP is predominantly synthesized in the liver as a pentamer and is upregulated in response to inflammatory cytokines, including interleukin (IL)-6 [88]. Additionally, CRP can undergo a structural change to expose neoepitopes [89], and under certain conditions such as increased urea or high temperature, it can irreversibly dissociate to its monomeric form (mCRP) [90,91]. It is speculated that, once the structural changes in CRP occur, it is able to exert its pro-inflammatory effects [87]. In this setting, mCRP has been shown to activate complement [92], endothelial cells [93,94], monocytes [95], neutrophils [96], and platelets [97,98], all of which have been identified as central in thrombus formation [53]. This understanding of the thromboinflammatory effects of mCRP has led to substantial interest in the evaluation of mCRP in cardiovascular disease. Indeed, mCRP has been shown to be deposited in infarcted myocardium and atherosclerotic plaques, whilst the therapeutic inhibition of CRP has been demonstrated to afford protection from myocardial ischaemia reperfusion injury [99]. Furthermore, mCRP has been localised on circulating microvesicles, combining two potential risk factors of VTE [100]. Overall, given its role in promoting thromboinflammation, it is tempting to speculate that mCRP may play an active role in promoting VTE, and therefore serve as a useful surrogate of VTE risk. However, to date, this remains to be investigated.

Nevertheless, the measurement of native CRP has been the subject of several prospective trials and appears to be an independent risk factor for arterial thrombosis and cardiovascular disease [101]. To this end, the Justification for the Use of Statin in Prevention: An Intervention Trial Evaluating Rosuvastatin (JUPITER) trial demonstrated that the treatment of healthy individuals with rosuvastatin based solely on elevated CRP led to a reduced incidence of both arterial and venous reinforcing the important association between CRP and thrombosis [102,103]. Additionally, CRP has been evaluated as a biomarker of acute VTE, and several studies have found an association between elevated CRP and the risk of venous thrombosis, although this finding was not universal [104,105,106,107,108,109,110,111]. Importantly, an elevated CRP appears to be important in predicting subsequent VTE with some promising results in both the general population as well as patients with malignancy-associated VTE [112,113].

## 6. The ‘Omics’ and VTE

In recent years, the improved understanding of many disease states at a molecular level has rapidly increased due to the availability of technology platforms analysing the ‘omics.’ The term ‘omics’ refers to technologies such as genomics, transcriptomics, proteomics, metabolomics, and lipidomics and in many fields is leading to personalised medical care [114]. In particular, genomics has already led to significant advances in the understanding of disease pathogenesis. Other platforms, such as metabolomics, are still relatively new and are likely to continue to enhance our understanding of disease in the future. Below, we address how these additional tools may assist in the current field of VTE.

### 6.1. Genetic Landscape of VTE

For well over a half a century, it has been well recognised that genetics plays a substantial role in the development of venous thrombosis, leading to the commonly available testing of several hereditary thrombophilias. In more recent years, genome-wide association studies have sought to improve the understanding of the heritability of VTE; however, to date, the findings have ultimately not altered routine clinical care [115,116,117,118,119,120]. The hereditary thrombophilias that are known to be associated with the risk of VTE include deficiencies in the natural anticoagulants, including antithrombin [121], protein C [122], and protein S [123], as well as genetic mutations in the factor V Leiden [124] and the prothrombin genes [125]. However, despite the established association between thrombophilia and risk of VTE, there is no substantial evidence that the intensity or duration of anticoagulation for VTE should be influenced by the discovery of one of the recognised thrombophilias [126]. Consequently, the clinical utility of routine laboratory testing for hereditary thrombophilias is questionable. Indeed, recent guidelines suggest that testing should only be considered in specific clinical circumstances and not used in a shotgun approach for all patients with VTE [126,127]. More recently, the improved access to high-throughput sequencing technologies, such as next-generation sequencing (NGS), has generated interest in establishing if other novel genetic markers may prove more useful in predicting the risk of recurrent VTE.

In this regard, Simeoni et al. evaluated the use of NGS using the ThromboGenomics platform for diagnosing thrombotic, bleeding, and platelet disorders. Covering 63 genes linked to these heritable conditions, they were able to show that the platform was effective at obtaining molecular diagnoses in patients with suspected thrombotic and bleeding disorders [128]. More recently, a 55-gene thrombophilia panel using whole-exome sequencing (WES) was performed on patients with acute VTE assessing genes associated with coagulation factors, natural anticoagulants, platelet function, vWF. The panel identified probable disease-causing variants or variants of unknown significance (VUS) in 60.9% of patients and was superior to current laboratory-based testing in identification. Furthermore, several studies have evaluated single-nucleotide polymorphisms (SNPs) in conjunction with clinical risk factors and assessed the utility of predicting acute VTE and have shown that the addition of SNPs to clinical factors may improve risk prediction [129,130].

To date, it seems we have yet to fully elucidate the role that NGS may play in the diagnosis of acute or recurrent venous thrombosis, and its utility as a tool in predicting recurrent VTE is yet to be determined. Additionally, the increasing use of genomics is accompanied by its own challenges, including adequate genetic counselling and the finding of VUS, where the clinical significance may not be fully understood. These issues should always be considered before proceeding with genetic testing in combination with ensuring appropriate pretest and posttest genetic counselling.

### 6.2. Proteomics

Proteomics refers to the study of protein quantification, structure, and localisation, coupled with characterisation of the interactions within a biological system, e.g., signalling pathways [131]. The study of the blood compartment proteome, including plasma, platelets, and leucocytes, has become an attractive tool in the field of VTE and has provided additional insight into the pathogenesis and molecular interactions taking place within the disease process. Several studies have evaluated proteomics to diagnose acute VTE as well as its utility in predicting recurrent disease. In the setting of acute VTE, plasma proteomic profiling has identified protein patterns that can be used to predict the diagnosis of acute VTE with relatively high specificity and have shown differences in proteins such as haptoglobin and alpha-1B glycoprotein, when compared with healthy controls [132,133]. Also, proteome analyses in urine of patients with acute VTE identified specific peptide patterns, which were further validated in both human and mouse tissue [134]. Additionally, prospective studies performing large-scale plasma proteomic profiling have identified predictive protein candidates, such as transthyretin, the vitamin K-dependent protein Z, and platelet-derived growth factor-beta (PDGFB), to have strong associations with the development of VTE. These novel biomarkers will require further validation but may be useful in predicting future, incident VTE [135,136].

Similarly, the proteome of other compartments, including MVs and platelets, has been evaluated in humans with VTE. Within MVs, two key proteins were enriched in patients with acute VTE—galactin-3 binding protein (Gal3BP) and alpha-2 macroglobulin. Interestingly, Gal3BP is involved in several pathways, including platelet activation and aggregation, upregulation of P-selectin, and promoting further MV shedding, and thus may be critical in propagating the early stages of venous thrombosis [137,138].

Additionally, studies of the platelet proteome have advanced the understanding of intracellular functional pathways, and the role that platelets play in thrombosis. Recently, the platelet proteome of patients with a positive lupus-anticoagulant (LA) has demonstrated several alterations when compared to healthy volunteers. One prominent finding is that protein disulphide isomerases are elevated in platelets of LA-positive patients, which are known to be involved with thrombus formation in vivo, further highlighting the importance of platelets in this disease process [139].

In summary, proteomics is a rapidly evolving field and continues to yield valuable information regarding the molecular pathways that are altered in venous thrombosis. At this stage, research that convincingly shows the potential of proteomics to predict recurrent VTE is limited; however, this field of research has the potential to unlock important biomarkers in the prediction of recurrent venous thrombosis.

### 6.3. Metabolomics and Lipidomics in VTE

The emerging fields of metabolomics and lipidomics have resulted in an increasing body of evidence suggesting that lipid species are vital to both procoagulant and anticoagulant effects within the plasma. Whilst our understanding of lipids in coagulation has traditionally centred on the role of PS as a source of phospholipid required for efficient thrombin generation [140], more recently, lipidomic approaches have identified several minor lipid species as important modulators of coagulation reactions. Notably, these studies have furthered the understanding that lipids have both procoagulant and anticoagulant effects in the plasma and have demonstrated a relationship between venous thrombosis and a change in lipid species [141,142,143,144].

Lipidomic analysis using mass spectrometry can evaluate hundreds of lipids in a single sample, which is a dramatic increase from conventional lipid analysis that focused primarily on cholesterol and triglyceride levels. This new technology has already unveiled plasma lipidome signatures that are predictive of cardiovascular disease risk and can be used in combination with traditional risk factors to enhance overall strategies to predict cardiovascular events and death [145,146]. More recently, the platelet lipidome has been established as a distinct compartment of lipid metabolism, with changes in the platelet lipidome found in patients with coronary artery disease [147,148].

Uncovering these alterations in the plasma and platelet lipidome in arterial thrombosis has further highlighted the importance of lipids in thrombosis, and it is hoped that novel approaches may also show promise in the prediction of venous thrombosis. In this regard, recent studies suggest that several lipid and metabolites pathways are altered in acute VTE [149,150,151]. Deguchi et al. have demonstrated an association between low plasma acylcarnitine levels and venous thrombosis and determined that acylcarnitines have anticoagulant activity by way of their ability to bind to, and inhibit, factor Xa, establishing a potential factor in the pathogenesis of venous thrombosis [144]. These novel findings add weight to the importance of lipid pathways in the formation of venous thrombosis, and our group are interested to see if these factors may be utilised in predicting recurrent disease.

## 7. Predicting VTE in Malignancy

The diagnosis of VTE is frequent in patients with malignancy, with an estimated incidence of up to 20% [152]. Moreover, treatment with chemotherapy has been shown to further increase the risk of developing venous thrombosis [153]. Importantly, patients diagnosed with malignancy-associated VTE are shown to have a worse prognosis compared to those without VTE, and both arterial and venous thrombosis are known to be among the leading causes of death in cancer patients [154,155]. The optimal management of malignancy-associated VTE presents a challenging scenario for clinicians as these patients have higher rates of bleeding when treated with anticoagulant therapy but also have an increased risk of recurrent thrombosis. These difficulties in treating malignancy-associated VTE have led to significant interest in identifying an ideal marker to prognosticate the risk of both initial and recurrent VTE in this population.

Due to the increased risk of VTE associated with malignancy, several risk prediction models have been evaluated in this setting, including the Khorana score [156], the Vienna Cancer and Thormbosis Study (CATS) score [157], and the PROTECHT score [158]. The most widely used prediction model is the Khorana score and it forms the basis for the other prediction models mentioned. The Khorana score uses the five variables (site of cancer, platelet count, haemoglobin, leucocyte count, and body mass index) to risk-stratify ambulatory outpatient patients for consideration of thromboprophylaxis. A recent systematic review and meta-analysis of studies using the Khorana score to risk-stratify patients with malignancy demonstrated that the score is certainly helpful in identifying patients at high risk of developing VTE. 

More recently, Pabinger et al. have developed a novel prediction model which utilises the variables of tumour-site category and D-dimer to differentiate between low and high risk of VTE in ambulatory cancer patients to give a predicted six-month risk of developing VTE [159]. This model will require further external validation in prospective studies but appears to improve upon previously available tools for predicting index VTE in this patient cohort. However, clinical prediction rules such as the Khorana score identify risk of index VTE during malignancy, rather than assessing risk of recurrence. Furthermore, the most commonly used clinical prediction rules actively exclude patients with malignancy due to the known prothrombotic phenotype associated with many types of malignancy and the higher risk of recurrent VTE. As such, many biomarkers have been studied for their role in predicting recurrence of malignancy-associated VTE, including D-dimer, sP-selectin, TF-positive MP, and CRP, and have been reviewed in depth elsewhere [160,161]. To date, no optimal marker has been found in predicting recurrent malignancy-associated VTE, and this is an area where further research is required to ensure that these patients with both challenging thrombotic and bleeding complications can receive the most appropriate treatment.

## 8. Limitations With the Use of Biomarkers in Predicting Recurrent VTE

Despite many recent advances in the field of VTE, the optimal biomarker to aid in predicting recurrent venous thrombosis is yet to be established and several limitations are recognised. With the exception of D-dimer, many of the biomarkers discussed above are not routinely measured in hospital laboratories and may only be available for assessment in a research setting. This may be costly for the patient and produce a result that is not validated for clinical decision-making. Additionally, several biomarkers, particularly MVs and EPCs, lack a standard definition, thus resulting in significant variation in classification between laboratories. Furthermore, despite rigorous quality control (QC) measures for laboratory accreditation, there remains variability between some commercial assays. An example of this is the use of D-dimer in the HERDOO2 rule, where it has been demonstrated that there is poor concordance between different commercial D-dimer assays leading to misclassification of risk prediction [17]. As such, differences in local practices, protocols, or assay use may result in substantial heterogeneity between biomarker results, and external validation procedures, coupled with thorough local laboratory QC measures, are of critical importance in this setting [162].

## 9. Future Perspectives and Conclusions

A considerable number of biomarkers have been evaluated for their potential utility in predicting episodes of recurrent venous thrombosis. To date, no single biomarker has been demonstrated to be superior to current clinical prediction models. However, recent studies have elucidated the key spatiotemporal events underpinning VTE pathogenesis and uncovered a remarkable array of novel factors that contribute to the development of venous thrombosis [53]. Importantly, these studies have highlighted that inflammation is central in the development of VTE and delineated novel molecular players in VTE pathogenesis. In this regard, neutrophil extracellular traps (NETs), which were first identified as a neutrophil response to bacterial stimulation, have been demonstrated to be present in human arterial and venous thrombi and appear to be critical activators of the intrinsic pathway in addition to serving as a scaffold for platelet adhesion and activation [163,164,165]. Similarly, platelets are increasingly recognised for their role in venous thrombosis. Platelets release a procoagulant inorganic polyphosphate, which activates FXII and appears to contribute to venous thrombosis in mice [166,167,168]. Additionally, platelets release the danger-associated molecular pattern (DAMP) protein, high mobility group box 1 (HMGB1), which serves to enhance monocyte recruitment and NET formation, both of which have been shown to be central to VTE formation in murine models [169,170,171]. Moreover, several proinflammatory cytokines and chemokines, including interferon-gamma, IL-6, and IL-17A, have been identified as being highly prothrombotic, whilst also playing an important role in leucocyte recruitment and activation; cementing the importance of the inflammatory response in venous thrombosis formation [172].

These novel molecular players remain to be investigated regarding their ability to aid in predicting recurrent VTE. However, our growing understanding of VTE pathogenesis, coupled with the rapid technical advances seen within the areas of genomics and lipidomics, raises the possibility that further functional novel biomarkers for recurrent VTE will soon be detected. Ultimately, we anticipate that the incorporation of both clinical factors and novel biomarkers will allow for more accurate prediction of VTE risk recurrence, thus ensuring that optimal treatment can be tailored to the individual, and subsequently improve patient outcomes in the field of venous thrombosis.

## Figures and Tables

**Table 1 jcm-09-01582-t001:** Comparison of clinical prediction models for recurrent venous thromboembolism.

Score	Variables	Inclusion Criteria	Definition of Unprovoked VTE	Findings—Risk of Recurrent VTE
DASH [11]	Abnormal D-dimer after AC Age ≤ 50 Gender Hormone therapy	First unprovoked VTE	**Absence of:** Surgery Trauma Active cancer Immobility Pregnancy and puerperium **Included:** Hormone therapy Thrombophilic blood abnormality (if no other VTE risks)	Annualised recurrence risk: Score ≤ 1:3.1% Score > 1:9.3%
HERDOO2 [8]	Gender Signs of post-thrombotic syndrome Abnormal D-dimer during on AC BMI ≥ 30 Age ≥ 65	First unprovoked VTE after 5–12 months of AC	**Absence of:** Major surgery within 3 months Malignancy within 5 years Immobilisation for ≥ 3 days Leg fracture or plaster cast **Included:** Travel-related Exogenous oestrogen Minor immobilisation Minor surgery	Annualised recurrence risk: Low-risk (0 or 1 factor) females: 1.6% High-risk (≥ 2 factors) females: 14.1% per year Males (no low-risk group identified): 13.7%
Vienna [9]	Gender VTE location D-dimer after ceasing AC	First unprovoked VTE after at least 3 months of AC	**Absence of:** Major surgery Major trauma Pregnancy Female hormone intake Hereditary thrombophilia Malignancy **Included:** Immobility	Continuous HR based on nomogram

BMI: body mass index; HR: hazard ratio; AC: anticoagulation; VTE: venous thromboembolism.

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
