# Peer review of "Predicting the Risk of Recurrent Venous Thromboembolism: Current Challenges and Future Opportunities"

_jcm, 2020, doi:10.3390/jcm9051582_

Round 1

Reviewer 1 Report

In this review the authors face a challenging issue in the field of venous thromboembolism (TEV) management, i.e. identification of strategies for the calculation of TEV recurrency.

The manuscript is well written and is a bird’s eye view on the topic. This review discusses clinical prediction models together with established and promising biomarkers that might be used to predict the development of recurrent VTE.

However, a major point that should be highlighted is a better differentiation between  established biomarkers and potentially useful biomarkers which still need confirmation and validation.

For instance, among biomarkers, d-dimer is considered similarly to P-selectin, microparticles and C-reactive protein.  D-dimer has been validated as predictor of VTE recurrence in solid studies and it is also included in all clinical prediction models that are discussed in the review. Even considering limits of d-dimer testing and highlighting cautions in interpreting the results, d-dimer cannot be considered a test to use in research setting. First, it is used in the clinical routine for its high negative predictive value, and this has not been demonstrated for p-selectin, CRP and microvesicles ; second, large majority of laboratories  can easily measure it by using standard coagulation instrumentation, whereas at least P-selectin and microvesicles require further instrumentation that might not be widely available in non specialized hospitals were VTE patients are managed.

In addition, 1 in 20 patients with normal d-dimer will develop recurrent VTE, whereas the same authors acknowledge that sP-selectin lacks sensitivity for recurrent VTE. Thus, the two biomarkers have significantly different impact as predictors of VTE recurrency.

Other points:

- Title: it should specified “predicting the risk of VTE recurrency”

- 4.1 In cancer patients it’s true that there are persisting risk factors requiring prolonged anticoagulation. However, there are patients with non active cancer requiring evaluation regarding duration of their anticoagulant therapy and in these patient finding of RVO might be of crucial relevance 

- 5.3 microvesicles, although interesting biomarker, is a test which suffers from many variables, as the authors themselves acknowledge. It should be highlighted that MV measurement is still preliminary for the prediction of VTE risk.

Author Response

In this review the authors face a challenging issue in the field of venous thromboembolism (TEV) management, i.e. identification of strategies for the calculation of TEV recurrency.

The manuscript is well written and is a bird’s eye view on the topic. This review discusses clinical prediction models together with established and promising biomarkers that might be used to predict the development of recurrent VTE.

However, a major point that should be highlighted is a better differentiation between  established biomarkers and potentially useful biomarkers which still need confirmation and validation.

For instance, among biomarkers, d-dimer is considered similarly to P-selectin, microparticles and C-reactive protein.  D-dimer has been validated as predictor of VTE recurrence in solid studies and it is also included in all clinical prediction models that are discussed in the review. Even considering limits of d-dimer testing and highlighting cautions in interpreting the results, d-dimer cannot be considered a test to use in research setting. First, it is used in the clinical routine for its high negative predictive value, and this has not been demonstrated for p-selectin, CRP and microvesicles ; second, large majority of laboratories  can easily measure it by using standard coagulation instrumentation, whereas at least P-selectin and microvesicles require further instrumentation that might not be widely available in non specialized hospitals were VTE patients are managed.

In addition, 1 in 20 patients with normal d-dimer will develop recurrent VTE, whereas the same authors acknowledge that sP-selectin lacks sensitivity for recurrent VTE. Thus, the two biomarkers have significantly different impact as predictors of VTE recurrency.

We thank the reviewer for this feedback. We agree that there is more data supporting D-dimer in this setting of recurrence when compared with other biomarkers. We have attempted to address this by adding additional information regarding the predicting value of D-dimer.  This has been addressed in section 5.1, lines 150, 151, 157-159

Other points

- Title: it should specified “predicting the risk of VTE recurrency”

Thank you – we have now added “recurrent” to title

- 4.1 In cancer patients it’s true that there are persisting risk factors requiring prolonged anticoagulation. However, there are patients with non active cancer requiring evaluation regarding duration of their anticoagulant therapy and in these patient finding of RVO might be of crucial relevance 

This is an interesting point, and we agree that this requires mentioning in the review. As such, we have addressed in section 4.1, lines 95 and 96, which now states “Of note, the one study population where RVO appears to be associated with recurrent VTE is in patients with malignancy-associated VTE, and RVO assessment may be useful when considering optimal duration of anticoagulation in this patient cohort but the overall number of patients evaluated has been small”

- 5.3 microvesicles, although interesting biomarker, is a test which suffers from many variables, as the authors themselves acknowledge. It should be highlighted that MV measurement is still preliminary for the prediction of VTE risk.

Thank you for this feedback and we agree that data is definitely preliminary. To this end, we have added the wording ‘remains preliminary to section 5.3 line 241.  Additionally, following feedback from all reviewers we have added section 8 addressing limitations with biomarkers in predicting recurrent VTE.

Reviewer 2 Report

In this paper, prediction models and biomarkers for predicting recurrent VTE are discussed, as well as challenges in this field. The paper reads well and discusses all relevant models. I have some remarks which could particularly improve the completeness of this review. In my view, the paragraph on biomarkers lacks some important factors. In addition, the problem of measurement heterogeneity needs to be discussed in order to inform readers that measuring biomarkers leads to new problems, which may therefore not always lead to better prediction models.

Line 68

Mention the name of the risk score here.

The utility of imaging in predicting recurrent VTE

It might be interesting to discuss the relatively new technique: Magnetic resonance direct thrombus imaging (MRDTI) and its role in predicting recurrent VTE.

Predictive performance of biomarkers

I would strongly advise to discuss the concept of measurement heterogeneity and its effect on predictive performance (see also: Luijken K et al. Stat Med. 2019 Aug 15;38(18):3444-3459. doi: 10.1002/sim.8183. Epub 2019 May 31.).

In VTE research for example, D-dimer can be measured with many assays. It was previously shown by developers of the HERDOO2 score that if other D-dimer assays are used than in the original HERDOO2 development study, predictive performance is attenuated which leads to unacceptable misclassification. (see: Rodger MA et al. Thromb Res. 2018 Sep;169:82-86. doi: 10.1016/j.thromres.2018.07.020. Epub 2018 Jul 17.)

How about the predictive performance of FVIII and other coagulation factors? (for example: Timp JF et al. J Thromb Haemost. 2015 Oct;13(10):1823-32 or Nemeth B et al. PLoS Med. 2015 Nov 10;12(11):e1001899; discussion e1001899. doi: 10.1371/journal.pmed.1001899.). In addition, it was recently shown that patients with high levels of circulating endothelial progenitor cells (EPC) are at low risk of recurrent VTE (EClinicalMedicine. 2019 Nov 27;17:100218. doi: 10.1016/j.eclinm.2019.11.011). In my opinion, this paragraph needs to be updated and extended to give a complete overview.

The title and abstract indicate this paper focusses on the prediction of recurrent VTE. Hence, in the paragraph on malignancy and VTE both models for predicting a first and recurrent VTE are discussed. This is somewhat confusing for the reader.

Finally, it might be worthwhile to add a closing paragraph summarizing clinical trials in which these prediction models have been applied to guide thromboprophylaxis treatment as there are some conflicting results on the effect of individualized thromboprophylaxis

Author Response

In this paper, prediction models and biomarkers for predicting recurrent VTE are discussed, as well as challenges in this field. The paper reads well and discusses all relevant models. I have some remarks which could particularly improve the completeness of this review. In my view, the paragraph on biomarkers lacks some important factors. In addition, the problem of measurement heterogeneity needs to be discussed in order to inform readers that measuring biomarkers leads to new problems, which may therefore not always lead to better prediction models.

Line 68

Mention the name of the risk score here.

  • Added – titled LTRRiP score

The utility of imaging in predicting recurrent VTE

It might be interesting to discuss the relatively new technique: Magnetic resonance direct thrombus imaging (MRDTI) and its role in predicting recurrent VTE.

  • Thank you for this suggestion and we definitely agree that this imaging technique should be addressed and included. As such, a paragraph evaluating MRDTI has been added into section 4.1, lines 99 – 108

Predictive performance of biomarkers

I would strongly advise to discuss the concept of measurement heterogeneity and its effect on predictive performance (see also: Luijken K et al. Stat Med. 2019 Aug 15;38(18):3444-3459. doi: 10.1002/sim.8183. Epub 2019 May 31.).

In VTE research for example, D-dimer can be measured with many assays. It was previously shown by developers of the HERDOO2 score that if other D-dimer assays are used than in the original HERDOO2 development study, predictive performance is attenuated which leads to unacceptable misclassification. (see: Rodger MA et al. Thromb Res. 2018 Sep;169:82-86. doi: 10.1016/j.thromres.2018.07.020. Epub 2018 Jul 17.)

  • We thank the reviewer for this suggestion, and we agree that the limitations surrounding the utility of biomarkers was not adequately addressed. As such, we have added section 8: Limitations with the use of biomarkers in predicting recurrent VTE to tackle these issues.

How about the predictive performance of FVIII and other coagulation factors? (for example: Timp JF et al. J Thromb Haemost. 2015 Oct;13(10):1823-32 or Nemeth B et al. PLoS Med. 2015 Nov 10;12(11):e1001899; discussion e1001899. doi: 10.1371/journal.pmed.1001899.) In addition, it was recently shown that patients with high levels of circulating endothelial progenitor cells (EPC) are at low risk of recurrent VTE (EClinicalMedicine. 2019 Nov 27;17:100218. doi: 10.1016/j.eclinm.2019.11.011). In my opinion, this paragraph needs to be updated and extended to give a complete overview.

  • Thank you for this feedback and additional suggestions in improving the scope of biomarkers reviewed in this article. In this regard, we have now included section 5.2, which evaluates coagulation factors, particularly factor FVIII in predicting recurrent VTE. Furthermore, we have added section 5.4. addressing the utility of endothelial progenitor cells.

The title and abstract indicate this paper focusses on the prediction of recurrent VTE. Hence, in the paragraph on malignancy and VTE both models for predicting a first and recurrent VTE are discussed. This is somewhat confusing for the reader.

We agree with the reviewer that the section evaluating VTE and malignancy, as well as VTE and bleeding were not completely in keeping with a review of biomarkers in recurrent VTE. To address this and sharpen the focus of the manuscript, we have added the word ‘recurrent’ to the title and additionally, we have added some information in section 7 - Predicting VTE in malignancy, lines 414 – 430 so ensure the differences in predicting index and recurrent VTE are further defined. Finally, we have decided to remove section 9 ‘Predicting risk of bleeding’ to reduce this confusion and focus on the topic of predicting recurrent VTE

Finally, it might be worthwhile to add a closing paragraph summarizing clinical trials in which these prediction models have been applied to guide thromboprophylaxis treatment as there are some conflicting results on the effect of individualized thromboprophylaxis

We thank the reviewer for this feedback. We have attempted to address the potential issues with clinical prediction models in section 3 and we have added further information (line 73 – 75) to demonstrate that there are conflicting results regarding these models. We note that there are newer prediction models to predict risk of VTE for hospitalised or immobilised patients, however to our knowledge these models look at acute VTE rather than recurrent. In this regard, we believe that this aspect may be outside the scope of this review.

Reviewer 3 Report

This is a well written and comprehensive review on the prediction of VTE. My only comment regards the lay out.

Initially, it looked like the authors were focussing on predicting recurrent VTE. Later also some room is given to predicting a first episode of VTE, i.e. in cancer patients. It would be wise to introduce the various topics early in the review.

Author Response

This is a well written and comprehensive review on the prediction of VTE. My only comment regards the lay out.

Initially, it looked like the authors were focussing on predicting recurrent VTE. Later also some room is given to predicting a first episode of VTE, i.e. in cancer patients. It would be wise to introduce the various topics early in the review.

Thank you for this feedback. We have added some additional clarification to section 7 ‘Predicting VTE in malignancy’ – lines 426 – 430, with the aim to focus on recurrent VTE. Furthermore, we have decided to remove section 9 ‘Predicting risk of bleeding on anticoagulation’ as we agree that this is outside the scope of the initials aims of the paper

Round 2

Reviewer 1 Report

No comment

Reviewer 2 Report

The authors adequately addressed my comments. In my view it is ready for publication.